# Predictors of Psychological Distress across Three Time Periods during the COVID-19 Pandemic in Poland

**DOI:** 10.3390/ijerph192215405

**Published:** 2022-11-21

**Authors:** Agata Chudzicka-Czupała, Soon-Kiat Chiang, Damian Grabowski, Marta Żywiołek-Szeja, Matthew Quek, Bartosz Pudełek, Kayla Teopiz, Roger Ho, Roger S. McIntyre

**Affiliations:** 1Faculty of Psychology, SWPS University of Social Sciences and Humanities, 40-326 Katowice, Poland; 2Department of Psychological Medicine, Yong Loo Lin School of Medicine, National University of Singapore, Singapore 119228, Singapore; 3School of Medicine, University College Dublin, Belfield, 4 Dublin, Ireland; 4Mood Disorders Psychopharmacology Unit, University Health Network, Toronto, ON M5T 2S8, Canada; 5Department of Psychiatry and Pharmacology, University of Toronto, Toronto, ON M5S 2E8, Canada; 6Brain and Cognition Discovery Foundation, Toronto, ON M4W 3W4, Canada; 7Institute for Health Innovation and Technology (iHealthtech), National University of Singapore, Singapore 117599, Singapore

**Keywords:** depression, anxiety, public health, COVID-19, pandemic, stress, policy, psychological impact, social determinants, fatigue

## Abstract

Background: Since the onset of COVID-19, public health policies and public opinions changed from stringent preventive measures against spread of COVID-19 to policies accommodating life with continued, diminished risk for contracting COVID-19. Poland is a country that demonstrated severe psychological impact and negative mental health. The study aims to examine psychological impact and changes in levels of depression, anxiety, and stress among three cross-sectional samples of Polish people and COVID-19-related factors associated with adverse mental health. Methods: In total, 2324 Polish persons participated in repeated cross-sectional studies across three surveys: Survey 1 (22 to 26 March 2020), Survey 2 (21 October to 3 December 2020), and Survey 3 (3 November to 10 December 2021). Participants completed an online survey, including Impact of Event Scale-Revised (IES-R), Depression, Anxiety, and Stress Scale (DASS-21), demographics, knowledge, and concerns of COVID-19 and precautionary measures. Results: A significant reduction of IES-R scores was seen across surveys, while DASS-21 scores were significantly higher in Survey 2. There was significant reduction in the frequency of following COVID-19 news, recent COVID-19 testing, and home isolation from Survey 1 to 3. Being emale was significantly associated with higher IES-R and DASS-21 scores in Surveys 1 and 2. Student status was significantly associated with higher DASS-21 across surveys. Chills, myalgia, and fatigue were significantly associated with high IES-R or DASS-21 scores across surveys. Frequency of wearing masks and perception that mask could reduce risk of COVID-19 were significantly associated with higher IES-R and DASS-21 scores. Conclusion: Conclusions: The aforementioned findings indicate a reduction in the level of the measured subjective distress andin the frequency of checking COVID-19 news-related information across three periods during the pandemic in Poland.

## 1. Introduction

Throughout human history, respiratory epidemics often result in a significant psychological morbidity in the general population, despite infection status [1]. Relatively high rates of anxiety, depression, and post-traumatic stress symptoms were reported in the general population during the COVID-19 pandemic in various countries [2]. Governments that implemented stringent measures to contain the spread of COVID-19 may be associated with subsequent effects on mental health [3]. Numerous lockdowns and different restrictive measures across governments heightened the psychosocial impact worldwide during the early stage of the COVID-19 pandemic [4]. Moreover, significantly higher risk of COVID-19-related hospitalization and death were reported in people with pre-existing depression [5].

Research conducted during the COVID-19 pandemic in Poland has clearly indicated a deterioration in the mental health of Polish residents, with a reported increase in the incidence of depression, anxiety, stress, and post-traumatic stress [6,7,8]. A previous study reported that Poland was one of the countries with the highest severity of psychiatric symptoms across three continents [9]. As the COVID-19 pandemic evolved, there was an increase in measures of somatization, fatigue, insomnia, loneliness, functioning impairment, and life dissatisfaction among Polish residents [10,11,12,13,14,15,16,17]. The increased incidence of symptoms of mental illness was reflected by an increased expenditure of psychiatric prescriptions, including antidepressants and hypnotics [18]. Twardowska-Staszek et al. (2021) reported that Polish people living in a medium-sized town or in a village is a predictor for negative emotion [19]. Recently, a gradual improvement in mental health has been reported, as Polish people were reported to be adapting to the “new normal” (i.e., public health policies that are less stringent in response to COVID-19 variants that cause less severe infection) [20]. To address persisting gaps in the research on the psychological impact of COVID-19 in Poland, a study is required to explore COVID-19-related factors associated with negative or decreased mental health in persons residing in Poland.

The primary aim of the study herein was to compare the psychological impact (i.e., the severity of depression, anxiety, and stress across three-time points using three cross-sectional surveys administered to Polish residents between 22 March to 26 March 2020, 21 October to 3 December 2020, and 3 November to 10 December 2021. The secondary aim of this study was to identify sociodemographic factors influencing the associations examined herein.

## 2. Methods

### 2.1. Study Design and Population

This study used the successive independent samples design where different samples of respondents from the population complete the survey over a time period. The successive independent samples design allows researchers to study changes in a population over time. The three waves of cross-sectional studies were conducted from 22 March to 26 March 2020 (Survey 1), 21 October to 3 December 2020 (Survey 2), and 3 November to 10 December 2021 (Survey 3). Survey 1 was conducted when Poland went through the first wave of COVID-19 pandemic throughout the country. As of 26 March 2020, the number of confirmed cases of COVID-19 infection was 1221, with 16 deaths reported in Poland [21]. Survey 2 was conducted during the second wave of the COVID-19 pandemic, with a rapid increase in new COVID-19 cases and related deaths. As of 3 December 2020, the number of confirmed cases and deaths rose to 14,838 confirmed cases and 620 deaths [21]. A rapid decline in COVID-19 cases and deaths were seen thereafter until 9 February 2021. Survey 3 was conducted during the third wave of COVID-19 pandemic in 2021, with a rapid increase in new COVID-19 cases and related deaths. As of 10 December 2021, the number of confirmed cases was 24,991 and the number of deaths was 571 [22]. Snowball sampling is a recruitment technique in which existing research participants were asked to assist the study team in identifying other potential research participants [23]. The snowball sampling strategy focused on recruiting participants from the general population living in various parts of Poland during the COVID-19 pandemic.

A total of 2324 individuals participated in three cross-sectional surveys, with 1064 participants for Survey 1, 557 participants for Survey 2, and 703 participants for Survey 3. Participants completed only one of three surveys (i.e., there are no repeat measures for a single participant).

### 2.2. Procedure

To comply with the social distancing and lockdown measures imposed by the Polish government, potential participants were invited to participate electronically. Information about this study and the survey was posted on social media (e.g., Facebook, LinkedIn) and on a website created by SWPS University. Participants were also encouraged to invite new participants from their contacts. The survey was delivered via two online survey platforms (i.e., Google Forms Online Survey on social media and SWPS University of Social Sciences and Humanities SONA platform). The Institutional Review Board of SWPS University, Poland, granted ethics approval for this study (WKEB62/04/2020). Informed consent was obtained from all participants and research data were anonymized and stored confidentially.

### 2.3. Outcomes

The study adapted and modified the National University of Singapore COVID-19 questionnaire [24]. The questionnaire consisted of questions related to (1) demographic data; (2) physical health status, health services contact, and contact history with COVID-19 in the past 14 days; (3) knowledge and concerns about COVID-19, and (4) precautionary measures against COVID-19. The Impact of Event Scale-Revised (IES-R) was used to measure the psychological impact of the COVID-19 pandemic [25]. The total IES-R score was divided into 0–23 (normal), 24–32 (mild psychological impact), 33–36 (moderate psychological impact), and >37 (severe psychological impact) [24] The Depression, Anxiety, and Stress Scale (DASS-21) was used to measure the levels of anxiety, depression, and stress of the participants [26]. For DASS-21, questions 3, 5, 10, 13, 16, 17, and 21 formed the depression subscale. The total depression subscale score was divided into normal (0–9), mild depression (10–12), moderate depression (13–20), severe depression (21–27), and extremely severe depression (28–42). Questions 2, 4, 7, 9, 15, 19, and 20 formed the anxiety subscale. The total anxiety subscale score was divided into normal (0–6), mild anxiety (7–9), moderate anxiety (10–14), severe anxiety (15–19), and extremely severe anxiety (20–42). Questions 1, 6, 8, 11, 12, 14, and 18 formed the stress subscale. The total stress subscale score was divided into normal (0–10), mild stress (11–18), moderate stress (19–26), severe stress (27–34), and extremely severe stress (35–42) [24]. Total DASS-21 score was used for analysis based on previous studies [27,28].

IES-R and DASS were used previously in various research related to the COVID-19 pandemic [29] and were validated in a Polish sample [30,31]. The Cronbach’s alpha for the Polish version of IES-R was 0.883. The Cronbach’s alpha for the Polish version of DASS-21 was listed as follows: DASS-21 stress: 0.890, DASS-21 anxiety: 0.854, DASS-21 depression: 0.886 [30].

### 2.4. Statistical Analysis

One-way analysis of variance (ANOVA) was used to compare the differences in mean IES-R and DASS-21 scores between Survey 1, 2, and 3. The Bonferroni correction was used when performing multiple comparisons between the IES-R and DASS scores for the three surveys. The categorical variables were presented as percentage of responses to the survey questions, which were calculated based on the number of participants per response out of the total possible responses to the question. Linear regression was used to calculate the univariate associations between the independent (e.g., health parameters, concerns about the COVID-19 pandemic) and dependent variables (e.g., IES-R and DASS-21 score) for the three surveys separately. The statistical tests were all two-tailed and with a significance level of *p* < 0.05. The statistical analysis was conducted by using SPSS Statistic 28.0.

## 3. Results

### 3.1. Comparison of Participants and Mental Health Status between the Three Surveys

Appendix A shows the comparison of the mean scores of DASS-21 stress, anxiety, and depression subscales and the IES-R scores between three surveys. The mean score (standard deviation, SD) for the DASS score was 15.85 (12.6) for participants in Survey 1, 20.60 (14.4) for the participants in Survey 2, and 17.61 (13.2) for participants in Survey 3.

The one-way ANOVA revealed that there was a statistically significant difference between at least two survey groups [F (2, 2321) = 23.6, *p* < 0.001]. The Bonferroni test for multiple comparison indicated that the mean value of DASS-21 score was significantly different between Survey 1 and 2 [*p* < 0.001, 95% C.l. = −6.41 to −3.10]; Survey 1 and 3 [*p* = 0.018, 95% C.I. = −3.31 to −0.23]; and Survey 2 and 3 [*p* < 0.001, 95% C.I. = 1.19 to 4.79]. The mean IES-R scores of participants in Survey 1 [31.19 (13.6)] and Survey 2 [30.04 (13.8)] were significantly higher than participants in Survey 3 [25.92 (13.7)]. The one-way ANOVA revealed that there was a statistically significant difference between at least two groups [F (2, 2321) = 32.5, *p* < 0.001]. The Bonferroni test for multiple comparisons indicated that the mean values of IES-R score were significantly different between Survey 1 and 3 [*p* < 0.001, 95% C.I. = −0.56 to 2.87] and between Survey 2 and 3 [*p* < 0.001, 95% C.I. = 2.26 to 5.98]. There was no statistical difference in IES-R score between Survey 1 and 2 [*p* = 0.322].

### 3.2. Demographic Characteristics and Their Association with Psychological Impact and Adverse Mental Health Status

The majority of the participants in Survey 1 were women (76%), middle aged ranging from 31 to 40 years (45.6%), married (55.5%), had a household size of 3–5 people (57.4%), were employed (84.68%), well-educated (73.1%) (i.e., having attained a bachelor degree or higher), and lived in a city/town (82.3%). Similarly, the majority of the participants in Survey 2 were women (75.9%), of the younger age group of 22 to 30 years (37.9%), single (69.8%), had a household size of 3–5 people (58.9%), were employed (61%), well-educated (54% with at least a bachelor degree), and lived in a city/town (80.6%). Likewise, the majority of the participants in Survey 3 were women (87.5%), single (72.7%), had a household size of 3–5 people (54.6%), were employed (64.4%), well-educated (50.1% with at least post-secondary school education), and lived in a city/town (87.8%).

The association between the demographic characteristics with IES-R scores and DASS-21 scores is presented in Table 1. Female sex was significantly associated with higher scores of IES-R and DASS-21 as compared to male participants in both Survey 1 and 2 (*p* < 0.001); however, this finding was not observed in Survey 3. Student status was significantly associated with higher DASS-21 scores (*p* < 0.05) as compared to employed participants in all three surveys. Significant association with higher DASS-21 scores was observed in participants with post-secondary school education (age 16–19 years) in Survey 2 and 3 (*p* < 0.001).

### 3.3. Physical Symptoms, Health Status, and Their Association with Psychological Impact and IES-R/DASS-21 Score

Findings related to physical symptoms and health status for the three surveys are shown in Appendix A. Fatigue (Survey 1: 28.3%, Survey 2: 45.6%, Survey 3: 52.1%), coryza (Survey 1: 25%, Survey 2: 29.3%, Survey 3: 23.6%), and sore throat (Survey 1: 19.2%, Survey 2: 18.9%, Survey 3: 18.3%) were the three most common physical symptoms reported by Polish participants. Approximately one quarter of participants consulted doctors in the past 14 days (Survey 1: 23% Survey 2: 27.5%, Survey 3: 29.4%). Hospitalization and recent quarantine in the past 14 days were uncommon (<5% in three surveys). There was a significant reduction in the recent COVID-19 testing from Survey 1 (23%) to Survey 3 (6.4%) (*p* < 0.001). The majority of participants reported good health status in three surveys (>80%). Contact with confirmed (23.3%) or suspected cases (32.7%) of COVID-19 infection and travelling to high-risk countries (7.4%) were significantly higher during Survey 2 as compared to other surveys (*p* < 0.001).

Physical health status and its association with the psychological parameters are presented in Table 2. Three physical symptoms, including chills, myalgia, fatigue, and poor self-rating health status were associated with either higher IES-R or DASS-21 scores in the three surveys (*p* < 0.05). Participants from Survey 1 and 3 who had consultation with a doctor in the past 14 days were significantly associated with higher IES-R and DASS scores (*p* < 0.01).

### 3.4. Knowledge and Concerns about COVID-19 and Their Association with Psychological Impact and IES-R/DASS-21 Score

Appendix A shows the comparison between the three survey participants on their knowledge of the transmission of COVID-19 and their concerns. In the three surveys, participants viewed transmission by droplets as the most common route of transmission (Survey 1: 99.2%; Survey 2: 98.9%; Survey 3: 97%) and transmission by food as the least common route of transmission (Survey 1: 16.3%; Survey 2: 15.6%; Survey 3: 17.2%). There was a significant reduction in satisfaction with health information from Survey 1 (44%) to Survey 3 (35.1%) (*p* < 0.001) and frequency in checking information about the pandemic from Survey 2 (10.2% who checked several times a day) to Survey 3 (0.7% who checked several times a day) (*p* < 0.001). Similarly, there was a significant reduction in the proportion of participants following COVID-19 news from other countries (Survey 1 61.7%; Survey 2: 27.3%; Survey 3: 16.5%) (*p* < 0.001). Interestingly, there was a significant reduction in the proportion of participants who were concerned about the economic impact (Survey 1: 49.8%, Survey 3: 38.8%) (*p* < 0.001), unemployment (Survey 1: 21.9%, Survey 3:15.9%) (*p* = 0.008), and extension of the COVID-19 pandemic (Survey 1: 62.6%, Survey 3: 46.1%) (*p* < 0.001). Furthermore, there was a significant reduction in concerns about incorrect diagnosis of COVID-19 (Survey 1: 68.5%; Survey 3: 21.3%) (*p* < 0.001).

Participants’ knowledge about COVID-19 transmission, their concerns, and their association with the psychological parameters are presented in Table 3. Participants who checked information regarding the status of the COVID-19 pandemic several times a day showed a significant association with higher IES-R and DASS scores. Concerns about lack of healthcare, own health status as well as family members’ COVID-19 status, and the likelihood of survival if infected with COVID-19 demonstrated a significant association with higher IES-R and DASS scores across all three surveys (*p* < 0.05).

### 3.5. Precautionary Measures for COVID-19 and Their Association with Psychological Impact and IES-R/DASS-21 Score

A comparison of the precautionary measures adopted by the participants is shown in Appendix A. There was a significant increase in the proportion of participants who agreed to wear a mask and protective gloves (Survey 1: 28.5%, Survey 2: 90.8%; Survey 3; 85.2%) (*p* < 0.001) and to wear a mask regardless of the presence or absence of symptoms (Survey 1: 34.9%, Survey 2: 93.2%, Survey 3: 87.9%) (*p* < 0.001). More than 50% of participants had a high level of belief in the effectiveness of mask as a protective measure in Surveys 2 and 3. COVID-19 vaccination rate was only reported in Survey 3, which was 79.5%.

In contrast, there was a significant reduction in the proportion of participants who isolated themselves at home (Survey 1: 78.3%; Survey 2: 35.5%; Survey 3: 28.3%) and practiced social distancing (Survey 1: 76.8%, Survey 2: 49.9%; Survey 3: 27.9%) (*p* < 0.001). There was a significant reduction in the proportion of participants who spent 20–24 h at home per day only (Survey 1: 65.3%; Survey 2: 39.5%; Survey 3: 27.3%).

Table 4 shows the association between precautionary measures and the psychological parameters in three surveys. Wearing a face mask and protective gloves (*p* < 0.05), covering mouth when coughing and sneezing (*p* < 0.05), washing hands with soap and water (*p* < 0.05), using disinfectants (*p* < 0.05), and social distancing (*p* < 0.05) were significantly associated with higher IES-R and DASS-21 scores in Survey 2 and 3. Vaccination against COVID-19 (*p* < 0.05), intention to receive vaccination (*p* < 0.05), and positive attitude towards vaccination (*p* < 0.05) were significantly associated with higher IES-R and DASS-21 scores in Survey 3. Similarly, participants who wore a mask regardless of the presence or absence of symptoms and were convinced about the effectiveness of masks were associated with higher IES-R and DASS-21 scores in Survey 3 (*p* < 0.01).

## 4. Discussion

This study aimed to compare the psychological status and predictors across three periods of the COVID-19 pandemic in Poland. The key findings are summarized as follows. For severity of psychiatric symptoms and psychological impact, there was a significant reduction in the IES-R score from Survey 1 to Survey 3, suggesting less psychological impact of the COVID-19 pandemic as it evolved. For demographic factors, female sex was significantly associated with higher IES-R and DASS-21 scores in Survey 1 and 2, while student status was significantly associated with higher DASS scores in three surveys. This foregoing finding replicates and extends other lines of research indicating that females (especially younger in age, i.e., <35 years) are at greater risk of psychological distress and mental health consequences during COVID-19 [32]. Regarding physical symptoms, chills, myalgia, and fatigue demonstrated a significant association with high IES-R or DASS-21 scores in three surveys. For health information and news, there was a significant reduction in the frequency of following COVID-19 news, recent COVID-19 testing, and home isolation from Survey 1 to Survey 3. The frequency of checking exhibited a positive and significant association with higher IES-R and DASS-21 scores. For precautionary measures, there was a significant increase in the proportion of participants who agreed to wear a mask from Survey 1 to Survey 3, although the frequency of wearing a mask and perception that wearing a face mask could reduce the risk of COVID-19 spread was significantly associated with higher IES-R and DASS-21 scores.

We found a significant reduction in IES-R score from Survey 1 to Survey 3, suggesting a lower psychological impact of the COVID-19 pandemic as it evolved. Our findings correspond to a recent three-wave study on Polish university students that found stress levels were significantly lower in the second and third wave of the COVID-19 pandemic [33]. During the three periods, frequency of checking COVID-19 news, concerns about the economic impact, unemployment and prolonging of the COVID-19 pandemic, social isolation, and social distancing were significantly higher during Survey 1 and significantly decreased in the subsequent periods (Survey 2 and 3). The above findings are in accordance with the Polish government’s removal of specific restrictions, orders, and prohibitions in relation to the state of the pandemic in early 2022 [34].

For factors associated with higher IES-R or DASS-21 scores, we found that the female sex was a significant risk factor for higher psychological impact and DASS-21 scores in Surveys 1 and 2. This finding is consistent with previous studies that found female sex was associated with psychological impact, depression, anxiety, or stress during the COVID-19 pandemic in China [24], Iran [35], Poland [33], Spain [28], and the United States [36]. The above findings suggest that healthcare practitioners should be more alert to the negative psychological impact on Polish women as the COVID-19 pandemic still evolves. We also found that student status was significantly associated with higher DASS-21 scores in three surveys. This finding is expected as the COVID-19 pandemic caused a major disruption of public examination that might affect the opportunities to enter universities [37]. This might explain why those participants with post-secondary school education reported a significant association with higher DASS-21 scores in Surveys 2 and 3. Previous research had identified specific factors, including exercise frequency, school reopening, self-quarantine or quarantine of classmates, taking temperature routinely, wearing masks routinely, sleep quality, cancellation of holiday, lockdown restriction, closure of several areas in school due to COVID-19, living conditions in the school, and taking the final examinations after school re-opening, as the primary influence factors for anxiety or depression in college students [38]. The education authority in Poland may consider strengthening online learning and examination to prepare for future pandemic and develop mental health strategy that is specially designed for students and focus on psychological resilience, coping strategy, and social support during the COVID-19 pandemic [39]. Other factors, such as concerns about lack of healthcare, personal health status, likelihood of survival if infected with COVID-19, and health status of family members if infected with COVID-19, were significantly associated with higher IES-R and DASS-21 scores in the three surveys.

A previous report found that physical symptoms resembling COVID-19 infection affected mental health status in the general population [9]. This study found that chills, myalgia, and fatigue were the three most significant physical symptoms associated with higher DASS-21 scores in the three surveys. Mosiolek et al. (2021) reported the co-occurrence of physical and psychiatric symptoms in people who suffer from COVID-19 infection [40]. Furthermore, Polish people who reported poor self-rating health status were significantly associated with higher DASS-21 scores in three surveys. Healthcare practitioners should pay attention to the above three physical symptoms and their association with adverse mental health. Views towards vaccination were unavailable in Survey 1 and 2 but vaccination was associated with higher IES-R and DASS-21 scores. The vaccination rate during Survey 3 was 79.5% and it was high. Similarly, psychiatric patients who suffered from anxiety and depression also demonstrated high acceptance of the COVID-19 vaccine [41]. For precautionary measures, wearing masks and gloves, hand hygiene, and social distancing were significantly associated with higher IES-R and DASS-21 scores in Survey 2 and 3. It is interesting to note that participants who were convinced that facemasks were an effective measure to reduce the risk of COVID-19 transmission were significantly associated with higher IES-R and DASS-21 scores during Survey 3. Previous research found cultural differences in acceptance of the use of face masks during the COVID-19 pandemic, with Europeans being less receptive [28,30]. Previous studies found that higher openness, conscientiousness, and neuroticism were associated with willingness to use COVID-19 precautionary behaviors [42,43]. Further research is required to study the relationship between personality traits and adherence to COVID-19 precautionary behaviors. This study has limitations that should be considered when interpreting the findings. First, due to the online and random recruitment, the participants who participated in the three surveys were predominantly women with a high level of education and who lived in a city or town. The study team tried their best to obtain the most representative sample of the Polish population, especially concerning the number of participants and their demographic characteristics under the COVID-19 restrictive measures. The nonprobability sampling limits the ability to generalize the results of the survey to the broader population.

As a result of the selection bias, the finding of this study could not be generalized to Polish people who are males, with lower levels of education and living in rural areas. Second, we inherited limitations as other repeated cross-sectional studies conducted in Poland [33] and China [37]. Participants from three surveys were different people and the random response to online recruitment could not allow repeated measures for the same or matched participants. This might affect the understanding of the causality between the COVID-19 pandemic and mental health in Poles. Third, this study mainly used self-reported questionnaires to measure psychiatric symptoms and did not make a clinical diagnosis. The gold standard for establishing psychiatric diagnosis involved a structured clinical interview and functional neuroimaging [44,45]. Objective diagnostic methods should be applied in future face-to-face research after COVID-19 restrictions are removed. Future studies should include in-depth qualitative interviews to identify other themes not reported in this study. Finally, we did not have access to the available pre-pandemic data before the pandemic that would allow a comparison of mental health parameters between and during the COVID-19 pandemic in Poland. Nevertheless, studies conducted in Poland indicate that the condition of mental health and levels of perceived stress have worsened compared to the pre-pandemic norm [12,46].

## 5. Conclusions

In conclusion, as Polish people adapted to living with COVID, there was a significant reduction in IES-R scores, following COVID-19 news, recent COVID-19 testing, and home isolation from Survey 1 to 3. Across three surveys, female sex, student status, and physical symptoms, such as chills, myalgia, and fatigue, demonstrated significant association with high DASS-21 scores. Although there was a significant increase in the proportion of Polish people who agreed to wear face masks from Survey 1 to 3, the frequency of wearing a face mask was significantly associated with higher IES-R and DASS-21 scores across three surveys. Taken together, the results of our analysis further underscore the mental health consequences of COVID-19 and invite the need for longer-term surveillance of the mental health of persons in Poland, in those with and without prior COVID-19 infection [47].

## Figures and Tables

**Table 1 ijerph-19-15405-t001:** Association between demographic variables and the psychological impact as well as adverse mental health status during the first, second and third surveys (n = 2324).

Demographic Variables	The First Survey (22–26 March 2020) (n = 1064)				The Second Survey (21 October–3 December 2020) (n = 557)				The Third Survey (3 November–10 December 2021) (n = 703)			
	Impact of Event		DASS (Stress, Anxiety or Depression Subscale		Impact of Event		DASS (Stress, Anxiety or Depression Subscale)		Impact of Event		DASS (Stress, Anxiety or Depression Subscale)	
	B	*p*-Value	B	*p*-Value	B	*p*-Value	B	*p*-Value	B	*p*-Value	B	*p*-Value
Gender Male	−0.80 ***	<0.001	−0.86 ***	<0.001	−0.85 ***	<0.001	–0.66 ***	<0.001	–0.35	0.099	–0.24	0.227
Female	Reference		Reference		Reference		Reference		Reference		Reference	
Age range												
12–21 years	−0.76	0.532	0.23 *	0.042	−0.95	0.110	0.83	0.164	0.59	0.235	1.50 **	0.003
22–30 years	−0.83	0.621	−0.17 **	0.014	−0.97	0.100	0.60	0.307	0.17	0.735	0.97	0.051
31–40 years	−0.56	0.477	−0.23 *	0.041	−0.92	0.130	0.12	0.844	0.23	0.654	0.70	0.170
41–49 years	−0.63	0.412	−0.28 *	0.035	−0.91	0.136	0.93	0.879	0.21	0.691	0.48	0.363
50–59 years	−0.45	0.978	−0.12	0.286	−0.72	0.288	0.35	0.603	Reference		Reference	
Above 60 years	Reference		Reference		Reference		Reference		NA		NA	
Marital status												
Married	0.03	0.964	−0.33	0.550	0.06	0.940	–1.09	0.165	−1.33	0.096	−1.09	0.378
Single	−0.13	0.820	−0.33	0.542	0.04	0.963	–0.61	0.432	−1.32	0.105	−0.63	0.606
Widowed	Reference		Reference		Reference		Reference		Reference		Reference	
Household Size												
6 people or more	0.30	0.821	−1.30	0.300	−0.28	0.882	−0.53	0.762	−0.58	0.483	−1.40	0.075
3–5 people	0.37	0.773	−1.21	0.323	0.21	0.906	−0.88	0.612	−0.12	0.867	−0.36	0.585
2 people	0.36	0.783	−1.33	0.279	0.31	0.864	−0.48	0.782	−0.41	0.559	−0.76	0.254
Staying alone	0.18	0.887	−1.40	0.256	0.49	0.790	−0.57	0.744	−0.09	0.901	−0.48	0.483
No one	Reference		Reference		Reference		Reference		Reference		Reference	
Employment status												
Unemployed	0.73 *	0.009	0.42	0.106	0.24	0.549	0.23	0.552	−0.48	0.132	−0.35	0.271
Retired	0.82 *	0.023	0.22	0.511	0.82	0.137	0.45	0.374	−0.53	0.665	0.35	0.778
Student	−0.10	0.612	0.50 *	0.011	−0.03	0.875	0.36 *	0.024	0.32 *	0.049	0.61 ***	<0.001
Employed	Reference		Reference		Reference		Reference		Reference		Reference	
Educational Level												
Primary school	−0.72	0.383	1.45	0.063	−0.13	0.844	0.83	0.184	0.43	0.683	0.56	0.575
Secondary school	−0.46	0.161	−0.53	0.092	0.08	0.845	0.45	0.276	−0.11	0.934	−0.23	0.854
Post-secondary school (19-21 years)	0.10	0.444	0.22	0.080	−0.04	0.823	0.60 ***	<0.001	0.15	0.273	0.64 ***	<0.001
University (Bachelor, Master, Doctorate)	Reference		Reference		Reference		Reference		Reference		Reference	
Residence												
City/Town	−0.04	0.789	−0.01	0.934	0.17	0.391	0.25	0.177	0.12	0.685	0.09	0.676
Village	Reference		Reference		Reference		Reference		Reference		Reference	

* *p* < 0.05, ** *p* < 0.01, *** *p* < 0.001.

**Table 2 ijerph-19-15405-t002:** **Association between physical health status and the psychological impact as well as adverse mental health status during the first, second and third surveys (n = 2324)**.

Physical Symptoms and Health Status	The First Survey (22–26 March 2020) (n = 1064)				The Second Survey (21 October–3 December 2020) (n = 557)				The Third Survey (3 November–10 December 2021) (n = 703)			
	Impact of Event		DASS (Stress, Anxiety or Depression Subscale		Impact of Event		DASS (Stress, Anxiety or Depression Subscale		Impact of Event		DASS (Stress, Anxiety or Depression Subscale	
	B	*p*-Value	B	*p*-Value	B	*p*-Value	B	*p*-Value	B	*p*-Value	B	*p*-Value
Physical Health Status												
Fever Yes	0.40	0.172	0.20	0.484	0.11	0.659	0.18	0.486	0.03	0.923	0.50	0.075
No	Reference		Reference		Reference		Reference		Reference		Reference	
Cough												
Yes	0.31 *	0.042	0.56 ***	<0.001	0.45 *	0.011	0.55 **	0.001	0.07	0.688	0.20	0.231
No	Reference		Reference		Reference		Reference		Reference		Reference	
Chills												
Yes	0.97 **	0.005	0.99 **	0.002	0.44	0.153	0.76 **	0.008	0.41	0.153	0.75 **	0.006
No	Reference		Reference		Reference		Reference		Reference		Reference	
Myalgia												
Yes	0.45 *	0.046	0.84 ***	<0.001	0.22	0.293	0.58 **	0.004	0.48 **	0.009	0.74 ***	<0.001
No	Reference		Reference		Reference		Reference		Reference		Reference	
Breathing difficulty												
Yes	0.78 *	0.014	1.16 ***	<0.001	0.67 *	0.011	1.04 ***	<0.001	0.13	0.658	0.56	0.051
No	Reference		Reference		Reference		Reference		Reference		Reference	
Coryza												
Yes	0.05	0.745	0.23	0.062	0.02	0.913	0.19	0.248	−0.05	0.753	0.50 **	0.001
No	Reference		Reference		Reference		Reference		Reference		Reference	
Sore throat												
Yes	0.31 *	0.027	0.49 ***	<0.001	0.18	0.372	0.36	0.055	0.16	0.373	0.56 ***	<0.001
No	Reference		Reference		Reference		Reference		Reference		Reference	
Fatigue												
Yes	0.58 ***	<0.001	0.80 ***	<0.001	0.52 ***	<0.001	0.92 ***	<0.001	0.33 *	0.018	0.78 ***	<0.001
No	Reference		Reference		Reference		Reference		Reference		Reference	
No complains												
Yes	−0.43 ***	<0.001	−0.75 ***	<0.001	−0.52 **	0.004	−0.82 ***	<0.001	−0.21	0.151	−0.65 ***	<0.001
No	Reference		Reference		Reference		Reference		Reference		Reference	
Health Services Contact												
Consultation with doctor in the past 14 days												
Yes	0.22	0.102	0.30 *	0.019	0.29	0.091	0.23	0.167	0.45 **	0.003	0.48 ***	<0.001
No	Reference		Reference		Reference		Reference		Reference		Reference	
Recent hospitalization in the past 14 days												
Yes	0.12	0.786	0.29	0.481	0.27	0.553	−0.42	0.345	0.03	0.957	0.02	0.966
No	Reference		Reference		Reference		Reference		Reference		Reference	
Recent quarantine in the past 14 days												
Yes	−0.22	0.764	−1.01	0.155	0.29	0.318	0.46	0.097	0.57	0.160	0.72	0.061
No	Reference		Reference		Reference		Reference		Reference		Reference	
Recent testing for COVID-19 in the past 14 days												
Yes	0.21	0.106	0.29 *	0.021	0.29	0.305	0.24	0.367	0.31	0.262	0.30	0.265
No	Reference		Reference		Reference		Reference		Reference		Reference	
Current self-rating health status												
Poor/Very poor	1.45 *	0.022	1.80 *	0.035	0.89 *	0.012	1.74 ***	<0.001	0.13	0.719	1.41 *	0.032
Average	0.78 ***	<0.001	1.01 ***	<0.001	0.43	0.065	0.83 *	0.024	0.64 ***	<0.001	0.82 ***	<0.001
Good/Very good	Reference		Reference		Reference		Reference		Reference		Reference	
Chronic illness												
Yes	0.39 **	0.004	0.37 **	0.005	0.14	0.447	0.20	0.246	0.29	0.085	0.31	0.058
No	Reference		Reference		Reference		Reference		Reference		Reference	
Contact history with COVID-19 in the past 14 days Close contact with an individual with confirmed infection with COVID-19												
Yes	0.44	0.557	−0.41	0.561	−0.15	0.408	−0.38 *	0.030	−0.03	0.881	0.10	0.576
No	Reference		Reference		Reference		Reference		Reference		Reference	
Indirect contact with an individual with confirmed infection with COVID-19												
Yes	−0.29	0.527	−0.06	0.899	0.06	0.736	−0.01	0.939	0.42	0.121	0.58 *	0.025
No	Reference		Reference		Reference		Reference		Reference		Reference	
Contact with an individual with suspected COVID-19												
Yes	−0.19	0.372	−0.34	0.104	0.28	0.094	0.23	0.136	0.02	0.934	0.06	0.744
No	Reference		Reference		Reference		Reference		Reference		Reference	
Contact with infected material												
Yes	0.26	0.092	0.30 *	0.039	0.57 **	0.004	0.31	0.094	−0.33	0.322	0.19	0.526
No	Reference		Reference		Reference		Reference		Reference		Reference	
Travel to high-risk countries with COVID-19												
Yes	−0.58	0.322	−0.48	0.141	−0.32	0.091	−0.23	0.418	−0.49	0.404	−0.14	0.792
No	Reference		Reference		Reference		Reference		Reference		Reference	
No contact												
Yes	NA		NA		−0.18	0.275	−0.11	0.477	−0.01	0.926	−0.27 *	0.040
No					Reference		Reference		Reference		Reference	

* *p* < 0.05, ** *p* < 0.01, *** *p* < 0.001.

**Table 3 ijerph-19-15405-t003:** Association of knowledge and concerns related to COVID-19 and the psychological impact as well as adverse mental health status during the first, second and third surveys (n = 2324).

Knowledge and Concerns Related to COVID-19	The First Survey (22–26 March 2020) (n = 1064)				The Second Survey (21 October–3 December 2020) (n = 557)				The Third Survey (3 November–10 December 2021) (n = 703)			
	Impact of Event		DASS (Stress, Anxiety or Depression Subscale		Impact of Event		DASS (Stress, Anxiety or Depression Subscale		Impact of Event		DASS (Stress, Anxiety or Depression Subscale	
	B	*p*-Value	B	*p*-Value	B	*p*-Value	B	*p*-Value	B	*p*-Value	B	*p*-Value
Route of transmission Droplets												
Yes	−0.12	0.848	0.18	0.753	−0.35	0.640	−0.42	0.556	0.92 *	0.043	0.39	0.308
No	Reference		Reference		Reference		Reference		Reference		Reference	
Transmitted through touch with infected person												
Yes	0.15	0.172	0.12	0.265	0.15	0.318	0.03	0.849	0.18	0.213	0.14	0.276
No	Reference		Reference		Reference		Reference		Reference		Reference	
Contact with contaminated objects												
Yes	0.31	0.051	0.29 *	0.049	−0.04	0.809	0.15	0.371	0.23	0.120	0.13	0.354
No	Reference		Reference		Reference		Reference		Reference		Reference	
Contact with infected blood (e.g. mosquito bite)												
Yes	0.40	0.081	0.01	0.975	−0.33	0.351	−0.36	0.284	0.23	0.385	−0.03	0.897
No	Reference		Reference		Reference		Reference		Reference		Reference	
Transmitted through food												
Yes	0.11	0.484	0.13	0.375	−0.12	0.569	0.17	0.398	0.23	0.208	−0.02	0.926
No	Reference		Reference		Reference		Reference		Reference		Reference	
Do not know												
Yes	0.43	0.639	0.43	0.621	0.55	0.373	0.81	0.164	−0.46	0.237	−0.34	0.334
No	Reference		Reference		Reference		Reference		Reference		Reference	
Satisfaction with the amount of health information about COVID-19												
Satisfied	−0.13	0.288	−0.12	0.332	0.18	0.387	0.04	0.821	0.02	0.886	−0.26	0.086
Not satisfied	−0.09	0.556	−0.17	0.249	0.24	0.175	0.11	0.516	0.23	0.216	−0.13	0.442
Do not Know	Reference		Reference		Reference		Reference		Reference		Reference	
How often do you check information regarding the status of the coronavirus pandemic?												
Several times a day	NA		NA		1.33 ***	<0.001	1.28 ***	<0.001	0.43	0.604	0.32	0.678
Once a day					0.77 ***	<0.001	0.58 **	0.007	0.50 *	0.045	0.09	0.771
Once every few days					0.22	0.052	0.38	0.051	0.57 ***	<0.001	0.33	0.995
I do not check	Reference		Reference		Reference		Reference		Reference		Reference	
Do you know what to do if you suspect coronavirus infection?												
Yes	−0.23	0.301	−0.69 **	0.001	−0.30	0.200	−0.57 *	0.010	−0.03	0.948	0.77 *	0.034
No	−0.59	0.456	−0.94	0.579	−0.07	0.613	0.06	0.139	−0.14	0.874	1.55 *	0.025
Difficult to say	Reference		Reference		Reference		Reference		Reference		Reference	
Do you follow the news from other countries regarding the development and course of the pandemic?												
Yes	0.53 ***	<0.001	0.26 *	0.025	0.28	0.125	0.06	0.713	0.16	0.418	−0.18	0.346
No	−0.08	0.767	−0.47	0.084	−0.47 *	0.017	−0.56 **	0.003	−0.38 *	0.016	−0.24	0.094
Occasionally	Reference		Reference		Reference		Reference		Reference		Reference	
Concerns about COVID-19 epidemic Concerns about lack of healthcare in case of infection with COVID-19												
Yes	0.49 ***	<0.001	0.46 ***	<0.001	0.51 **	0.002	0.40 *	0.011	0.57 ***	<0.001	0.36 **	0.006
No	Reference		Reference		Reference		Reference		Reference		Reference	
Concerns about own health status if infected with COVID-19												
Yes	0.57 ***	<0.001	0.42 ***	<0.001	0.75 ***	<0.001	0.59 ***	<0.001	0.72 ***	<0.001	0.66 ***	<0.001
No	Reference		Reference		Reference		Reference		Reference		Reference	
Concerns about health status of family members if infected with COVID-19												
Yes	0.42 **	0.007	0.53 ***	<0.001	0.54 **	0.003	0.45 **	0.010	0.55 ***	<0.001	0.62 ***	<0.001
No	Reference		Reference		Reference		Reference		Reference		Reference	
Concerns about likelihood of surviving if infected with COVID-19												
Yes	0.78 ***	<0.001	0.78 ***	<0.001	0.49 **	0.007	0.53 **	0.002	0.72 ***	<0.001	0.62 ***	<0.001
No	Reference		Reference		Reference		Reference		Reference		Reference	
Concerns about economic impacts of coronavirus												
Yes	−0.08	0.483	0.04	0.723	0.31 *	0.044	0.22	0.135	0.33 *	0.019	0.29 *	0.031
No	Reference		Reference		Reference		Reference		Reference		Reference	
Concerns about losing a job												
Yes	0.35 **	0.009	0.45 ***	<0.001	0.15	0.432	−0.05	0.788	0.34	0.068	0.37 *	0.041
No	Reference		Reference		Reference		Reference		Reference		Reference	
Concerns about incorrect diagnosis of COVID-19												
Yes	0.31 **	0.009	0.17	0.140	0.08	0.682	0.07	0.681	0.48 **	0.004	0.34 *	0.032
No	Reference		Reference		Reference		Reference		Reference		Reference	
Concerns about extended epidemic duration												
Yes	0.35 **	0.003	0.19	0.091	0.01	0.961	−0.11	0.473	0.43 **	0.002	0.27 *	0.037
No	Reference		Reference		Reference		Reference		Reference		Reference	
Concerns about re-lockdown												
Yes	NA		NA		NA		NA		0.08	0.588	0.05	0.688
No									Reference		Reference	
Concerns of lack of vaccine against coronavirus												
Yes	NA		NA		NA		NA		0.30	0.327	0.39	0.188
No									Reference		Reference	
No concerns												
Yes	−1.41 ***	<0.001	−1.92 ***	<0.001	−1.42 **	0.004	−1.50 ***	<0.001	−1.14 ***	<0.001	−1.45 ***	<0.001
No	Reference		Reference		Reference		Reference		Reference		Reference	

* *p* < 0.05, ** *p* < 0.01, *** *p* < 0.001.

**Table 4 ijerph-19-15405-t004:** Association of precautionary measures related to COVID-19 and the psychological impact as well as adverse mental health status during the first, second and third surveys (n = 2324).

Precautionary Measures	The First Survey (22–26 March 2020) (n = 1064)				The Second Survey (21 October–3 December 2020) (n = 557)				The Third Survey (3 November–10 December 2021) (n = 703)				
	Impact of Event		DASS (Stress, Anxiety or Depression Subscale		Impact of Event		DASS (Stress, Anxiety or Depression Subscale		Impact of Event			DASS (Stress, Anxiety or Depression Subscale	
	B	*p*-Value	B	*p*-Value	B	*p*-Value	B	*p*-Value	B	*p*-Value		B	*p*-Value
Wearing mask and protective gloves													
Yes	0.17	0.158	0.10	0.398	0.54 *	0.047	0.90 ***	<0.001	0.43 *	0.032		0.60 **	0.001
No	Reference		Reference		Reference		Reference		Reference			Reference	
Covering mouth when coughing and sneezing													
Yes	−0.02	0.851	−0.13	0.292	0.39 *	0.028	0.77 ***	<0.001	0.55 **	0.003		0.43 *	0.011
No	Reference		Reference		Reference		Reference		Reference			Reference	
Washing hand with soap and water													
Yes	−0.40	0.141	−0.54 *	0.034	0.83 ***	<0.001	0.41	0.063	0.65 **	0.004		0.68 ***	<0.001
No	Reference		Reference		Reference		Reference		Reference			Reference	
Using disinfectants													
Yes	0.02	0.867	−0.22	0.056	0.41 *	0.019	0.47 **	0.005	0.51 ***	<0.001		0.46 ***	<0.001
No	Reference		Reference		Reference		Reference		Reference			Reference	
Self-isolating at home													
Yes	0.10	0.468	0.23	0.079	0.38 *	0.018	0.72 ***	<0.001	0.25	0.109		0.25	0.091
No	Reference		Reference		Reference		Reference		Reference			Reference	
Avoiding touching nose, mouth and eyes													
Yes	0.07	0.529	−0.07	0.544	NA		NA		NA			NA	
No	Reference		Reference										
Eating healthy													
Yes	−0.18	0.107	−0.46 ***	<0.001	−0.06	0.703	-0.39 **	0.008	−0.05	0.702		−0.41 **	0.002
No	Reference		Reference		Reference		Reference		Reference			Reference	
Social distancing													
Yes	0.15	0.246	−0.01 ***	<0.001	0.48 **	0.002	0.39 **	0.010	0.58 ***	<0.001		0.09	0.532
No	Reference		Reference		Reference		Reference		Reference			Reference	
Not applicable													
Yes	−0.01	0.981	0.46	0.289	−0.71	0.134	−1.38 **	0.002	−0.52	0.201		−0.84 *	0.023
No	Reference		Reference		Reference		Reference		Reference			Reference	
I am vaccinated against COVID-19													
Yes	NA		NA		NA		NA		0.37 *	0.037		0.65 ***	<0.001
No									Reference			Reference	
I intend to get vaccinated again soon													
Yes	NA		NA		NA		NA		0.39 **	0.008		0.43 **	0.002
No									Reference			Reference	
I’m going to get vaccinated for the first time soon													
Yes	NA		NA		NA		NA		0.08	0.868		0.59	0.182
No									Reference			Reference	
Attitude towards vaccination against coronavirus													
Positive and vaccinating									0.77 **	0.001		0.43 *	0.045
Positive but not vaccinating									0.85 *	0.017		0.12	0.704
Negative	NA		NA		NA		NA		0.08	0.825		−0.67 *	0.039
Difficult to say									Reference			Reference	
Wearing a mask regardless of the presence or absence of symptoms													
Yes	0.31 **	0.008	0.26 *	0.021	0.59	0.055	0.72 *	0.014	0.75 ***	<0.001		0.98 ***	<0.001
No	Reference		Reference		Reference		Reference		Reference			Reference	
Are you convinced of the need to wear a mask?													
Yes	NA		NA		0.23	0.170	0.60 ***	<0.001	0.57 ***	<0.001		0.39 **	0.005
No					Reference		Reference		Reference			Reference	
Are you convinced and how much are you convinced about the effectiveness of the mask as a protective measure													
Fully convinced					0.20	0.466	0.60 *	0.021	0.90 ***	<0.001		0.74 **	0.002
Fairly convinced					0.43	0.115	0.77 **	0.003	0.93 ***	<0.001		0.89 ***	<0.001
Somewhat convinced	NA		NA		−0.20	0.471	0.30	0.262	0.73 **	0.005		0.81 ***	<0.001
Fairly unconvinced					−0.04	0.914	0.16	0.604	0.36	0.205		0.56 *	0.030
Completely unconvinced					Reference		Reference		Reference			Reference	
Average number of hours staying at home per day to avoid COVID-19													
20–24 h	0.04	0.832	−0.03	0.871	−0.14	0.492	−0.50 *	0.013	−0.06	0.741		−0.09	0.625
10–19 h	−0.31 *	0.025	−0.26 *	0.034	−0.33	0.371	−0.38 *	0.024	0.11	0.311		0.19	0.085
0–9 h	Reference		Reference		Reference		Reference		Reference			Reference	

* *p* < 0.05, ** *p* < 0.01, *** *p* < 0.001.

## Data Availability

The data presented in this study are available on request from the corresponding author.

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
