# Peer review of "Predictors of Psychological Distress across Three Time Periods during the COVID-19 Pandemic in Poland"

_ijerph, 2022, doi:10.3390/ijerph192215405_

Round 1

Reviewer 1 Report

Dear Authors,

I find this article interesting, but I wish that the sampling could have been done in a way to obtain a representative sample of Polish population, especially concerning the number of participants and their demographic characteristics. I believe you should have tried to obtain a more representative sample by including other subpopulations (male, rural, elderly, without computer skills, etc.) using phone interview, prepaid mail, and/or other means.

In the Abstract section the conclusion is not evident from the results and I suggest it should be rewritten.

In the Methods section the snowball sampling strategy should have a description and/or a reference.

Discussion lacks comparison between obtained results of mental health outcomes and general Polish population results before the COVID-19 pandemic. Also, it lacks outlined limitation regarding determining causality of findings in cross-sectional studies. 

Grammar check and style correction are needed.

With kind regards.

Reviewer 2 Report

1) I cannot see Figure 1 described in line 126.

2) It is not convenient to read the text if you must look once in table in the main text and next in supplementary table, and again, and again.

3) There is not a table with characteristics of samples, described in lines 146-155, the first table should contain characteristics and should be desribed on the beginning of the Results section as the first subsection.

4) lines 123-125 information about sample size and unpaired samples should be presented earlier before Statistical methods.

5) line 117 Regression - what was modelled? what dependent variable, what independent variables? Are variables numerical or categorical?

6) Subsection 2.3 - Please write more about scales, for example if higher scores mean higher depression etc. What intervals for none, mild, moderate and severe depression, anxiety and stress are? Check DASS-21 Scoring Instruction.  If all items are in the same direction or we must reverse. The same for IES-R.

7) in abstract and in results it is not clear, women had higher depression than men etc. The desription of the results is not satisfied. What does "DASS higher" mean? the same for IES-R.

8) In many reference, DASS-21 is separately described for depression, anxiety and stress. not total score.

9) only one decimal place is enought for integer numbers.

10) in tables 1,2,3 better write p instead of t

11) lines 131,138 - it is not clear, difference in what?

12) it is not clear "adverce mental helath status" for ex. in title 3.5

13) suppl table 3 - it is enougt write Yes with n (%), when no+yes=100%.

Reviewer 3 Report

Hello

Thank you for your fine work

I have some minor correct suggestions

Table 1 in the third line from above: keep t or T

Line 178: Will you consider:" The majority" instead of "major"

I dont find any explanation in lines 300-305 why those 3 symptoms should alert family doctors and refer to mental health help. The statistical significant association in this case should be further explained and the authors should look for an answer and suggest an hypothesis at least. To conclude from those specific results that doctors should refer patients presenting symptoms of chills, malaise and fatigue to mental health, is not serious recomendation. This point is underdeveloped. You can't just make statistics a reason for such recomendation without any real discussion
